# Voltage-Gated Sodium Channels as Potential Biomarkers and Therapeutic Targets for Epithelial Ovarian Cancer

**DOI:** 10.3390/cancers13215437

**Published:** 2021-10-29

**Authors:** Iris S. Brummelhuis, Stephen J. Fiascone, Kathleen T. Hasselblatt, Gyorgy Frendl, Kevin M. Elias

**Affiliations:** 1Faculty of Health, Medicine, and Life Sciences, Maastricht University, 6200 MD Maastricht, The Netherlands; iris.brummelhuis@radboudumc.nl; 2Surgical ICU Translational Research Center, Brigham and Women’s Hospital, Boston, MA 02115, USA; gfrendl@bwh.harvard.edu; 3Division of Gynecologic Oncology, Department of Obstetrics, Gynecology, and Reproductive Biology, Brigham and Women’s Hospital, Boston, MA 02115, USA; Stephen.Fiascone@bmc.org (S.J.F.); khasselblatt@bwh.harvard.edu (K.T.H.); 4Dana-Farber Cancer Institute, Boston, MA 02115, USA; 5Harvard Medical School, Boston, MA 02115, USA; 6Department of Anesthesiology, Perioperative and Pain Medicine, Brigham and Women’s Hospital, Boston, MA 02115, USA

**Keywords:** high-grade serous ovarian cancer, sodium channels, local anesthetics

## Abstract

**Simple Summary:**

Voltage-gated sodium channels are membrane proteins that change conformation in response to depolarization of the membrane potential, allowing sodium ions to flow into cells. While voltage-gated sodium channels are normally studied in terms of neuron impulses and skeletal or cardiac muscle contraction, abnormal ion channel expression is a feature of many cancer cells. The aim of our study was to assess the expression of voltage-gated sodium channels in ovarian cancer cells. We found that ovarian cancer cells generally express lower levels of voltage-gated sodium channels than normal cells and that two voltage-gated sodium channels, *SCN8A* and *SCN1B*, were prognostic biomarkers for ovarian cancer overall survival. In vitro studies suggested that drugs that block voltage-gated sodium channels, such as certain anti-epileptic drugs and local anesthetics, might sensitize ovarian cancer cells to chemotherapy. These findings suggest that voltage-gated sodium channels may be interesting targets for ovarian cancer therapy.

**Abstract:**

Abnormal ion channel expression distinguishes several types of carcinoma. Here, we explore the relationship between voltage-gated sodium channels (VGSC) and epithelial ovarian cancer (EOC). We find that EOC cell lines express most VGSC, but at lower levels than fallopian tube secretory epithelial cells (the cells of origin for most EOC) or control fibroblasts. Among patient tumor samples, lower *SCN8A* expression was associated with improved overall survival (OS) (median 111 vs. 52 months; HR 2.04 95% CI: 1.21–3.44; *p* = 0.007), while lower *SCN1B* expression was associated with poorer OS (median 45 vs. 56 months; HR 0.69 95% CI 0.54–0.87; *p* = 0.002). VGSC blockade using either anti-epileptic drugs or local anesthetics (LA) decreased the proliferation of cancer cells. LA increased cell line sensitivity to platinum and taxane chemotherapies. While lidocaine had similar additive effects with chemotherapy among EOC cells and fibroblasts, bupivacaine showed a more pronounced impact on EOC than fibroblasts when combined with either carboplatin (ΔAUC −37% vs. −16%, *p* = 0.003) or paclitaxel (ΔAUC −37% vs. −22%, *p* = 0.02). Together, these data suggest VGSC are prognostic biomarkers in EOC and may inform new targets for therapy.

## 1. Introduction

Epithelial ovarian cancer (EOC) is the leading cause of gynecologic cancer death among women in developed countries [1]. EOC is treated with a combination of surgical cytoreduction and platinum-based chemotherapy. Primary remission rates are high, but most patients develop recurrence within three years, with an extremely low rate of salvage [2]. Despite this adverse prognosis, there remains a sizable minority of patients with EOC, even with advanced-stage disease on presentation, who achieve long-term survival rates exceeding ten years [3]. Understanding the factors that contribute to the more favorable prognoses in these cases has the potential to inform novel treatment strategies for the larger share of women with EOC.

Abnormal ion channel expression distinguishes several types of carcinoma, including breast, colon, pancreatic, prostate, lung, esophageal, and gastric cancers [4,5]. Ion channels not only have a role as prognostic biomarkers, but also as potential therapeutic targets using several classes of existing pharmacologic compounds [6]. Among the various channel types, voltage-gated sodium channels (VGSC) have received particular interest because VGSC activity appears to potentiate the invasiveness and metastatic potential of carcinoma cells [7,8,9,10]. Indeed, preclinical data suggest that VGSC blockade using either anti-epileptic drugs (AEDs) or local anesthetics (such as lidocaine or bupivacaine) may inhibit tumor growth and metastasis [11,12,13,14,15,16,17,18,19].

While few patients with EOC will be exposed to AEDs, many will receive local anesthetic agents. As with AEDs, local anesthetics appear to encourage tumor cell apoptosis and inhibit tumor migration. In the past, the translational relevance of this finding was limited because patients were primarily exposed only to small doses of local anesthetics via local wound infiltration. However, modern enhanced recovery after surgery (ERAS^®^) protocols have dramatically increased utilization of local anesthetics due to an emphasis on opioid-sparing perioperative management [20,21]. These techniques produce greater systemic exposure to local anesthetics, either from continuous bupivacaine or lidocaine neuraxial analgesia catheters (which can produce notable serum levels of drug) or continuous intravenous lidocaine infusions [22,23,24,25]. Whether these changes in analgesic management may affect cancer outcomes has been a topic of considerable debate, both for ovarian cancer as well as other malignancies [26,27,28,29,30,31,32,33,34,35]. As the role of VGSC channel expression in EOC has not been extensively studied, we sought to understand the prognostic value of VGSC expression in EOC tumors and to explore the effects of VGSC inhibition on EOC cells.

## 2. Materials and Methods

### 2.1. IRB Approval

This study was approved by Mass General Brigham Institutional Review Board under protocol #2016P002742.

### 2.2. RNAseq Data

RNAseq data regarding VGSC expression for 48 different ovarian cancer cell lines were downloaded from the Cancer Cell Line Encyclopedia (CCLE), a publicly available dataset from the Broad Institute (Cambridge, MA USA) [36,37]. RNAseq expression sets measured in log_2_ Robust Multiarray Average (RMA) units were downloaded from the CCLE webserver https://portals.broadinstitute.org/ccle/ (accessed on 7 January 2018). RNAseq data comparing VGSC expression between three EOC cell lines (KURAMOCHI, OVSAHO, and JHOS4) and three immortalized benign fallopian tube cell lines (FT33, FT194, and FT246) were downloaded in fragments per kilobase per million mapped reads (FPKM) from the Gene Expression Omnibus, accession number GSE83101 [38]. Sequence tags were mapped to the reference genome Hg19 using TopHat v2.0.6 (Johns Hopkins University, Baltimore, MD USA), and transcript levels were calculated in FPKM using Cufflinks v2.0.2 (Johns Hopkins University, Baltimore, MD USA). Differential expression was determined with CuffDiff, using Chi-square tests with 1 degree of freedom and two-tailed *p*-values to assess statistical significance [39]. Adjusted *p*-values were calculated using a Bonferroni correction for multiple testing. RNA-seq data appear in Appendix A.

### 2.3. Quantitative Reverse Transcription Polymerase Chain Reaction (qRT-PCR)

RNA was extracted from the EOC lines KURAMOCHI and OVCAR5 and the fibroblast cell lines BJ and WI-38 using TRIzol (Thermo Fisher Scientific, Waltham, MA, USA) and chloroform (Fisher Scientific, Fair Lawn, NJ, USA) then quantified using a NanoDrop (Thermo Fisher Scientific, Waltham, MA, USA). RNA quality was assessed by UV spectroscopy. RNA was then diluted to 40 ng/ul, and reverse transcription was performed using the TaqMan Reverse Transcription Kit (Thermo Fisher Scientific, Waltham, MA, USA) to generate cDNA. Quantitative PCR was performed using TaqMan Array Human Voltage-Gated Ion Channel plates (Thermo Fisher Scientific, Waltham, MA, USA). Three plates were run for each cell line using RNA extracted on three separate days, thus representing three biologic replicates. The Step One Plus Real Time PCR system (Applied Biosystems, Thermo Fisher Scientific, Waltham, MA, USA) was used for all quantitative PCR experiments. Boxplots were constructed using GenEx v.6.1 (MultiD Analyses AB, Göteborg, Sweden). Raw threshold cycle (Ct) values were loaded into GenEx v.6.1 (MultiD Analyses AB, Göteborg, Sweden) and normalized to expression of the reference genes GAPDH, HPRT1, and GUSB. Relative fold changes were calculated using the 2^−ΔΔCt^ method with fibroblast cell lines as the referent and transformed to a log_2_ scale. Raw Ct data, as well as normalized data for sodium channel expression, are provided in Appendix A and Appendix A, respectively.

### 2.4. Kaplan–Meier Analysis

Kaplan–Meier curves based on tumor VGSC expression were generated for overall survival (OS) using KMplot software, from a database of public microarray datasets derived from tumors of ovarian cancer patients downloaded from Gene Expression Omnibus and The Cancer Genome Atlas (http://kmplot.com/analysis) (accessed on 26 February 2018) [40]. Results were analyzed from 699 ovarian cancer patients (539 confirmed serous, 19 confirmed endometrioid, 141 unknown histology) known to have undergone optimal surgical cytoreduction (<1 cm residual disease) and received platinum-based chemotherapy. Data from outlier microarrays (as defined by the KMplot software) were excluded. The actual number of patients for analysis of each gene varied by the presence of the mRNA probes on the included microarrays. For each gene, the optimal microarray probe was selected according to the Jetset method described by Li, et al. [41]. Samples were split into high and low expression groups using a published method for determining the best performing threshold as a cutoff and compared using a log-rank test [42].

### 2.5. Cell Lines and Culture Conditions

The human high-grade serous ovarian cancer cell lines KURAMOCHI, OVCAR3, OVCAR5, JHOS4, and OVSAHO were a gift from Dr. Dipanjan Chowdhury (Dana-Farber Cancer Institute, Boston, MA, USA) and have been previously characterized [43,44]. Human fibroblast cell lines BJ and WI 38, human breast cancer cell line T47D, and human choriocarcinoma cell lines JEG3 and JAR were purchased from American Type Culture Collection (ATCC) (Manassas, VA, USA).

OVCAR3, OVCAR5, and T47D cells were routinely cultured with RPMI 1640 medium (ATCC) supplemented with 1% penicillin-streptomycin (ATCC) and 10% or, for OVCAR3, 20%, fetal bovine serum (FBS) (ATCC). The OVCAR3 and T47D culture medium additionally contained 0.01 units/mL and 0.02 units/mL insulin, respectively. KURAMOCHI and OVSAHO were routinely cultured with DMEM/F12 (ATCC) supplemented with 1% penicillin-streptomycin (ATCC) and 10% FBS (ATCC). JEG3, JAR, WI 38, and BJ were grown in Eagle’s Minimal Essential Medium (EMEM; ATCC) supplemented with 10% FBS plus 1% penicillin-streptomycin. Cells were grown in 10 cm diameter tissue culture dishes (BD Biosciences, San Jose, CA, USA) and were maintained in a humidified 5% CO_2_ atmosphere at 37 °C. Cells were plated for assays from culture plates in log growth phase, at approximately 70–90% confluence. All cells were tested negative for mycoplasma via a PCR assay (Charles River Laboratories, Wilmington, MA, USA). Cell line identities were verified by short tandem repeat (STR) testing using the commercially available PowerPlex^®^ 18D Kit (Promega, Madison, WI, USA).

### 2.6. Drugs

The local anesthetic agents lidocaine, bupivacaine, benzocaine, and procaine and the chemotherapeutic agents carboplatin and paclitaxel were purchased as salts from Sigma-Aldrich (St. Louis, MO, USA). The sodium channel inhibitors zonisamide and rufinamide were purchased as salts from Selleck Chemicals (Houston, TX, USA).

Stock solutions of all drugs were prepared by dissolving the drugs in sterile water, except for the stock solutions of benzocaine and paclitaxel, which were prepared by dissolving the drugs in ethanol. The stock solutions were aliquoted and stored at −20 °C. The stock solution concentrations were based on the manufacturer stated limits of solubility of the compounds. Immediately prior to use, final test concentrations were achieved by making a serial dilution of stock solutions with standard growth medium.

### 2.7. Cell Proliferation Assays

All cells were plated in triplicate in 96-well tissue culture plates (BD Biosciences) at a density of 2000 cells per well and allowed to attach overnight.

For construction of IC_50_ curves, cells were then incubated with a serial dilution range of bupivacaine (6 uM to 3.75 mM), lidocaine (0.2 to 125 mM), benzocaine (8 uM to 5 mM) or procaine (0.02 to 12.5 mM). An equivalent volume percent of ethanol was used as a control for possible changes to serum concentration within the wells. In all assays, standard growth medium alone was used as a negative control. Initially, cell proliferation was assessed at both 48-h and 96-h time points; however, it soon became clear that 48 h was sufficient to see drug effects, thus this time point was used for subsequent studies.

For studying the effect of local anesthetics combined with chemotherapy, cell lines were allowed to attach overnight, then first incubated in standard growth medium in the presence or absence of 10 mM lidocaine or 1 mM bupivacaine (these concentrations being near the half-maximal inhibitory concentration [IC_50_] for most of the cell lines) for 24 h. Then the medium was removed, and a logarithmic concentration range of chemotherapy was added. Cells were incubated with chemotherapy for 72 h.

To study the effect of AEDs, cells were treated with 50 uM or 500 uM rufinamide and zonisamide for 4 h. Standard growth medium was used as a negative control. Cells were incubated for 72 h prior to the MTT assay.

Cell proliferation was measured using a thiazolyl blue tetrazolium bromide assay (MTT; Sigma-Aldrich). After treatment, the medium was removed, and the cells were incubated with 20 uL MTT reagent (5 mg/mL; Sigma-Aldrich) in the dark for four hours. MTT is a yellow soluble salt that is converted to dark blue, insoluble formazan by mitochondrial oxidoreductases of viable cells. The formazan crystals were solubilized with acidified isopropanol (Thermo Fisher Scientific, Waltham, MA, USA) and quantified by absorbance spectrophotometry at a wavelength of 570 nm using a plate reader (Biotek, Winooski, VT, USA). The absorbance of the control group without treatment was defined as 100%, and the absorbance of the other groups was calculated relative to the control.

### 2.8. Statistical Analysis

All experiments were repeated at least three times. IC_50_ values were constructed using four-parameter nonlinear regression. If the predicted IC_50_ value was beyond the range of concentrations tested experimentally, then the IC_50_ was considered not reached. The area under the curve (AUC) was calculated using the trapezoid rule as a pharmacokinetic cumulative measurement of drug effect [45]. A lower IC_50_ or AUC indicates greater drug potency or cell line sensitivity, respectively. For statistical comparisons of IC_50_ values or AUC, Student’s *t*-test was used. For comparisons of groups, one-way ANOVA and two-way ANOVA tests were used where appropriate. An alpha value <0.05 was considered statistically significant. Analyses were performed using GraphPad Prism v.6 (San Diego, CA, USA). For qRT-PCR analysis, relative quantities of sodium channel expression between ovarian cancer lines and fibroblast cell lines were compared in GenEx v.6.1 (MultiD Analyses AB, Göteberg, Sweden) using a *t*-test with Dunn–Bonferroni correction for multiple testing.

## 3. Results

### 3.1. VGSC Expression in Ovarian Cancer Cell Lines

The Cancer Cell Line Encyclopedia contained VGSC RNAseq data on 48 ovarian cancer cell lines (Appendix A). Across histologic subtypes of ovarian cancer, most VGSC were expressed, except SCN10A. The most highly expressed VGSC were SCN1B and SCN8A (Figure 1A). To compare the relative expression of VGSC in EOC cells to normal tissues, we used a previously published RNAseq dataset (Appendix A) from our laboratory comparing EOC cells to benign fallopian tube secretory epithelial cells (FTSEC), believed to be the cells of origin for most EOC (Figure 1B). Interestingly, while confirmed as the two highest expressed VGSC among the EOC cell lines, both SCN1B (−864 fold; *p* < 0.001) and SCN8A (−21 fold; *p* < 0.001) were expressed at markedly lower levels in EOC compared to FTSEC. We considered the possibility that fallopian tube cells might have unusually high expression levels of these VGSC. Therefore, both to validate the RNAseq expression data and to test this hypothesis, we measured relative VGSC expression in the EOC cell lines OVCAR5 and KURAMOCHI compared to the fibroblast cell lines BJ and WI 38 by qRT-PCR (Figure 1C). Once again, compared to the benign cell lines, the EOC cell lines had significantly lower expression of the VGSC SCN8A (−12 fold) and SCN1B (−10 fold), as well as SCN2A (−259 fold) and SCN9A (−3113 fold) (all *p* < 0.001).

### 3.2. Prognostic Implications for VGSC Expression in Ovarian Cancer

Based on the cell line data, we focused on the relationship between SCN8A and SCN1B tumor expression and overall survival from EOC after optimal cytoreductive surgery followed by platinum-based chemotherapy. Lower expression of SCN8A was strongly associated with improved OS (median OS 111 months vs. 52 months; HR = 2.04 [1.21–3.44; *p* = 0.007]), while lower expression of SCN1B was associated with poorer OS (median OS 45 months vs. 56 months; HR = 0.69 [0.54–0.87; *p* = 0.002]) (Figure 2A,B).

### 3.3. Effects of AEDs on EOC Cells

The contrasting prognostic significance of SCN8A and SCN1B tumor expression raised the possibility that VGSC blockade could be either helpful or harmful EOC. To examine this further, we tested the effects of sodium channel blockade in vitro. We began by testing the effects of VGSC inhibition using AEDs. EOC cell lines were treated at either low (50 µM) or high (500 µM) concentrations using either of two AEDs, zonisamide and rufinamide, known to be semi-selective sodium channel inhibitors (Figure 3). At the higher concentrations, but not the lower concentrations, both drugs moderately inhibited EOC proliferation with an average change across the EOC cell lines of −13% ± 3.8% (*p* < 0.001) for zonisamide and −21% ±3.8% (*p* = 0.005) for rufinamide.

### 3.4. Effects of Local Anesthetics on Cell Proliferation

As noted above, local anesthetics are another class of VGSC inhibitors and are more likely to be given to EOC patients. Whereas AEDs prolong the inactivation state of VGSC by stabilizing the inactive state and preventing the return of the channels to the active form, local anesthetics stabilize the open state of the sodium channel, thus preventing depolarization. We tested the effects of two amide local anesthetics (lidocaine and bupivacaine) and two ester local anesthetics (benzocaine and procaine) using serial dilution curves. Several non-ovarian cancer cell lines were also included to test for the specificity of any observed effects. An isovolumetric titration curve of ethanol was used as a negative control with no significant toxicity seen (Appendix A).

The local anesthetics caused concentration-dependent decreases in cell proliferation in all the cells lines (Figure 4A–D). On balance, the magnitude of the effects was greater than those seen with the AEDs. IC_50_ values were calculated for each local anesthetic, except for benzocaine, where IC_50_ values were not reached (Appendix A). Across all cell lines, the mean IC_50_ value of bupivacaine (1.7 mM) was significantly lower than either lidocaine (6.2 mM; *p* < 0.001) or procaine (8.3 mM; *p* = 0.035). As bupivacaine and lidocaine are the most clinically relevant compounds for perioperative management, these were then selected for subsequent studies.

### 3.5. Impact of Local Anesthetic Exposure on Subsequent Response to Chemotherapy

We next investigated how modulating VGSC function might impact subsequent chemotherapy response. Ovarian cancer cell lines or fibroblast cell lines (as controls) were treated with a 24-h pulse of 10 mM lidocaine or 1mM bupivacaine (chosen from the IC_50_ curves). The media were then removed, and cells were treated with the two most commonly used drugs in ovarian cancer chemotherapy, carboplatin or paclitaxel, for 72 h.

The addition of local anesthetics shifted the cell proliferation curves downward for all the cell lines, resulting in lower AUC for the chemotherapeutic agents (Figure 5A–J; Figure 6A,B). However, while both cell types showed similar reductions in AUC after exposure to lidocaine (Figure 6C), the effect of bupivacaine was cell type-specific, as ovarian cancer cells showed much greater reductions in AUC after exposure to bupivacaine for both carboplatin (−1.02 [−37%] vs. −0.39 [−16%], *p* = 0.003) and paclitaxel (−0.82 [−37%] vs. −0.52 [22%], *p* = 0.02) compared to fibroblasts (Figure 6D).

## 4. Discussion

Aberrant ion channel expression is a hallmark of many different types of human cancers [6,46]. Depending on the cellular context, VGSC have been characterized as both potentiating or suppressing tumor growth [5,47,48,49]. In this report, we have identified that most VGSC are expressed in ovarian cancer cells, but at lower expression levels than in normal tissues. While individual VGSC can be associated with either improved or poorer prognoses for EOC patients, on balance it appears that inhibition of VGSC in EOC has a generally inhibitory effect on cell growth. In addition, these data suggest that exposure to local anesthetics may influence the subsequent effectiveness of chemotherapy in EOC and that this may be a cell type and drug-specific effect.

Our results are consistent with prior reports that have suggested that local anesthetics may directly influence cancer cell growth and response to chemotherapy [16,17,18,50,51,52]. For example, combined treatment with lidocaine and cisplatin promoted a higher level of breast cancer cell apoptosis than singular lidocaine or cisplatin treatment via demethylation of RARβ2 and RASSF1A [45]. In another study, for mice bearing leukemic cells, intraperitoneal administration of cisplatin combined with procaine on days 1 and 5 produced 33% and 50% cure rates, while the cure rates obtained with cisplatin alone were 17% and 9%, respectively [53]. Furthermore, novel complexes of platinum and lidocaine or platinum and procaine are both more active compounds against cancer cells than platinum alone [54,55]. However, making general conclusions from these studies has been difficult. Most of these studies reviewed one drug only with great variation in dose ranges (often several orders of magnitude) among studies. Additionally, most of these articles studied the effects on only one cell line, and only a few compared tumor cells to non-tumor cells.

Another study on local anesthetics and ovarian cancer, by Xuan, et al., supports our findings [52]. The authors found that clinically relevant concentrations of bupivacaine enhanced sensitivity to paclitaxel chemotherapy. Unfortunately, they described only one ovarian cancer cell line, and the cell line they used, SKOV3, has recently been classified as a poor ovarian cancer model [56]. Here, we have shown that this inhibitory effect of local anesthetics holds true across several ovarian cancer cell lines with higher fidelity to human tumors, and we have also added a prognostic role for VGSC using data from patient specimens [43,44].

While there are possible beneficial effects of combining chemotherapy and local anesthetics, interactions between VGSC and chemotherapy have also been linked to increased toxicities for both platinum and taxane therapies. Oxaliplatin causes apoptosis in neuronal cells by slowing ion currents across VGSC [57,58]. Moreover, genetic polymorphisms in VGSC have been associated with the risk of oxaliplatin-induced chronic peripheral neuropathy [59,60,61]. Similarly, VGSC are affected by the polymerization of tubulin. Paclitaxel, which enhances tubulin polymerization, impairs both slow and fast inactivation of VGSC [62]. These effects have been connected to the increased incidence of cardiac arrhythmias during taxane therapy [63].

Our study does have some important limitations. The patient data that we used were compiled from microarray gene expression profiles of tumor specimens. These are retrospective data, and we do not have extensive clinical information on these samples to understand whether the patients were exposed to local anesthetics during their therapy. In some studies, use of local anesthetics has been associated with improved EOC survival [28,64]. However, this benefit would presumably only extend to patients with higher VGSC tumor expression, which would be expected to dilute the effect of low VGSC as a positive prognostic marker. We also did not explore protein-level expression of VGSC. A protein biomarker which could be assessed by immunohistochemistry or Western blot would be much more useful for clinical application. Moreover, differences in mRNA may not always translate into protein expression differences. In future studies, we hope to assess the mechanism of VGSC inhibition in ovarian cancer more in-depth and plan to include protein level assessments in these investigations. Another limitation is that we do not know the individual effects of each VGSC. While this could be studied by knocking down specific VGSC in vitro, in practice, the clinical relevance for this would be limited, as drugs do not exist with this level of channel selectivity. However, development of channel-selective agents may be an important area for future research. To this end, it would also be interesting to examine individual cancer cell lines with extremes of response to VGSC inhibition to elucidate more precisely the mechanism underlying these differences. We hope to address this in future studies. In conclusion, expression of VGSC in ovarian tumors is prognostic of overall survival for EOC after optimal surgery and platinum-based chemotherapy. Targeting VGSC may augment responses to carboplatin and paclitaxel chemotherapy. If so, novel agents could offer a new adjunct to enhance current therapeutic strategies, and ion channels may be interesting prognostic biomarkers to include in clinical studies. These efforts must be undertaken carefully given the potential for VGSC to increase chemotherapy toxicity. Ultimately, understanding the ion channel expression of individual EOCs before cancer treatment may lead to more effective individualized/personalized anti-cancer regimens.

## Figures and Tables

**Figure 1 cancers-13-05437-f001:**
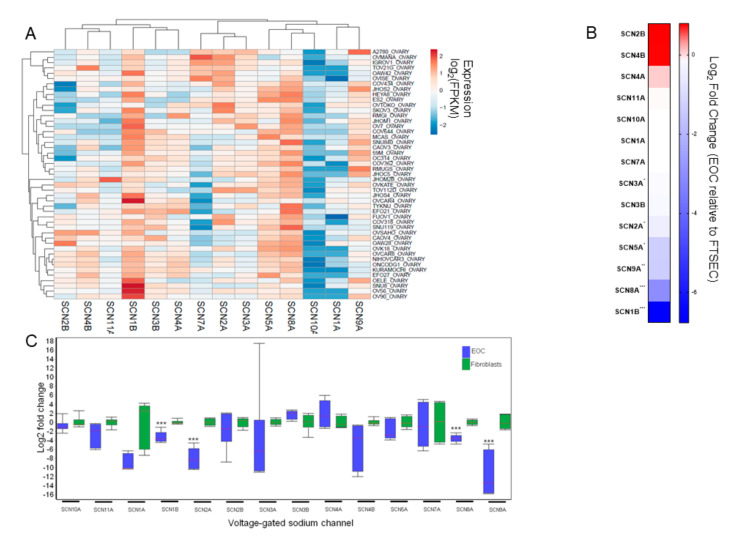
(**A**) Expression of voltage-gated sodium channels by RNAseq among 48 ovarian cancer cell lines in the Cancer Cell Line Encyclopedia. (**B**) Relative expression of voltage-gated sodium channels by RNAseq in epithelial ovarian cancer cells (EOC) compared to fallopian tube secretory epithelial cells (FTSEC). (**C**) Box and whisker plots showing relative expression of voltage-gated sodium channels by qRT-PCR in EOC versus fibroblasts. *** *p* < 0.001 Adjusted for multiple testing.

**Figure 2 cancers-13-05437-f002:**
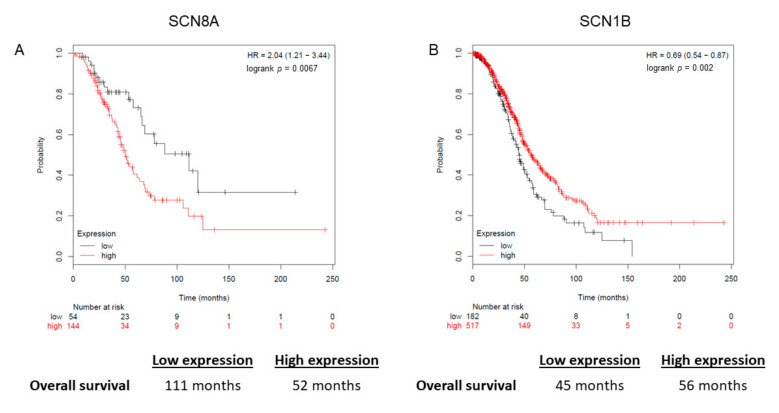
Overall survival after optimal cytoreductive surgery followed by platinum-based chemotherapy stratified by tumor voltage gated sodium channel expression of (**A**) SCN8A and (**B**) SCN1B.

**Figure 3 cancers-13-05437-f003:**
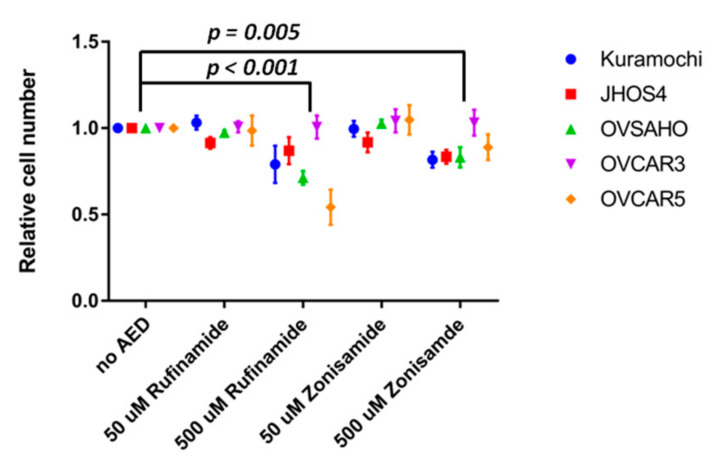
Effect of treatment with anti-epileptic drugs (AEDs) on epithelial ovarian cancer cell proliferation. Two-way ANOVA adjusted for multiple comparisons.

**Figure 4 cancers-13-05437-f004:**
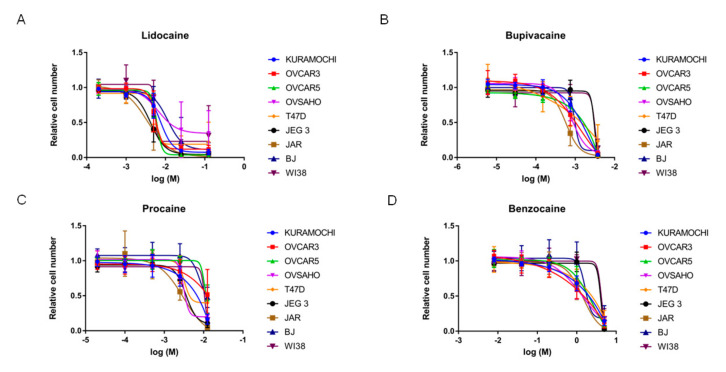
Concentration-dependent effects of local anesthetics on cell proliferation in various cell lines for (**A**) lidocaine, (**B**) bupivacaine, (**C**) procaine, and (**D**) benzocaine.

**Figure 5 cancers-13-05437-f005:**
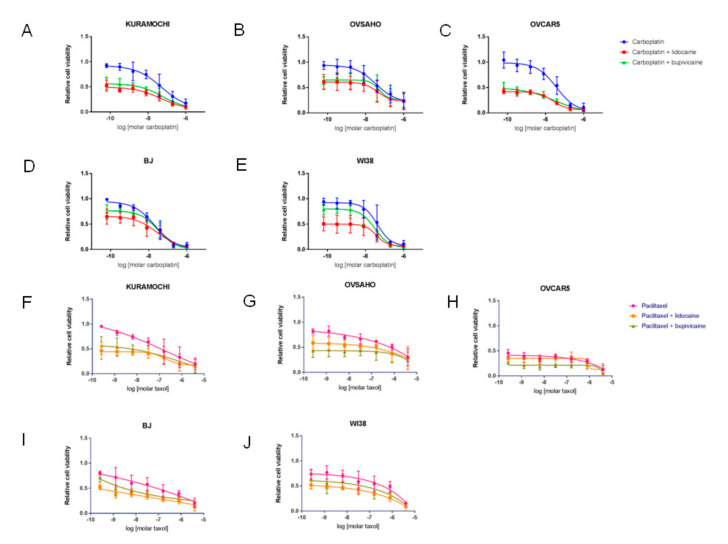
Effects of lidocaine and bupivacaine on ovarian cancer cell (KURAMOCHI, OVSAHO, OVCAR5) or fibroblast (BJ, WI38) responses to carboplatin treatment (**A**–**E**) or paclitaxel treatment (**F**–**J**).

**Figure 6 cancers-13-05437-f006:**
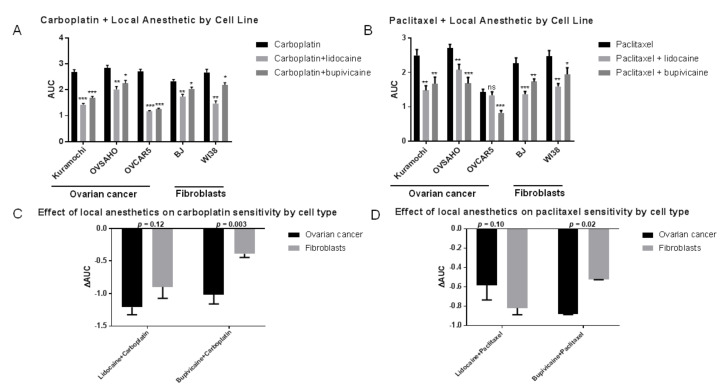
Change in AUC after addition of local anesthetic for (**A**) carboplatin and (**B**) paclitaxel by cell line. Comparisons for each cell line are relative to chemotherapy alone. Relative change in AUC between fibroblasts and ovarian cancer cells after addition of local anesthetic for (**C**) carboplatin and (**D**) paclitaxel. * *p* < 0.05, ** *p* < 0.01, *** *p* < 0.001, ns—not significant.

## Data Availability

The data presented in this study are available in the Appendix A. Additional data are available on request from the corresponding author.

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
