# Peer review of "Voltage-Gated Sodium Channels as Potential Biomarkers and Therapeutic Targets for Epithelial Ovarian Cancer"

_cancers, 2021, doi:10.3390/cancers13215437_

Round 1

Reviewer 1 Report

Brummelhuis et al., report evidence that VGSC expression in epithelial ovarian cancer can be used as a prognostic biomarker, as well as a therapeutic target.  VGSC RNA expression in human tumors correlates with aggressiveness.  Treatment with either anti-epileptic or local anesthetic drugs slows tumor progression. The manuscript is well written.  The appropriate statistics are used.

While there are many reports that various carcinomas overexpress VGSCs, suggesting that they may be useful as therapeutic targets, the authors are commended for highlighting their prognostic value.  However, this reviewer was disappointed that not all cell lines used to show efficacy of anti-epileptics and local anesthetics appear in the RNASeq table in Fig. 1.  For example, OVCAR 3 and 5 are used to show relative efficacy of rufinamide and zonisamide, but VGSC subtype expression is not reported.  Because OVCAR3 appears insensitive to these two drugs, whereas OVCAR5 appears sensitive, it would be useful to relate this finding to VGSC subtype expression in the two lines.  Perhaps we will see this in a future report.

Author Response

Thank you for this insight. The RNAseq data presented in Figure 1 were obtained from the Cancer Cell Line Encyclopedia (CCLE), a publicly available dataset. Unfortunately, expression data for VGSCs was not available for all of the cell lines used in this study. However, RNA-seq data for OVCAR3 does appear in Figure 1, just under a different name: NIH:OVCAR3. VGSC expression data for OVCAR5 appears in the supplemental dataset.  The study was not powered to look at differences in ovarian cancer cell lines relative to one another as much as to focus on cancer cell lines relative to normal cells, but we agree that selecting cancer cell lines at the extremes of response to AEDs and using this as a platform to focus on mechanisms of response in in vivo studies would an excellent idea for a future report. To this end, we have added the following statement to the discussion (lines 408-412):

“To this end, it would also be interesting to examine individual cancer cell lines with extremes of response to VGSC inhibition to elucidate more precisely the mechanism underlying these differences.  We hope to address this in future studies.”

Reviewer 2 Report

The Authors present a paper entitled “Voltage-gated sodium channels as potential biomarkers and therapeutic targets for epithelial ovarian cancer”. They have explored the relationship between voltage-gated sodium channels (VGSC) and epithelial ovarian cancer (EOC), finding that EOC cell lines express most VGSC, but at lower levels than fallopian tube secretory epithelial cells of origin for most EOC) or control fibroblasts.  As the role of VGSC channel expression in EOC has not been extensively studied, we sought to understand the prognostic value of VGSC expression in EOC tumors and to explore the effects of VGSC inhibition on EOC cells. The problem is well presented and the background sufficiently described in the Introduction.

The statistical analysis and the Materials and Methods description is appropriate. The work is well conceived and developed, even though all the protein analysis level is missing. I would strongly recommend that the authors analyze VGSC expression in the cell lines used for dose response experiments.

Author Response

Thank you for the review. We acknowledge that protein level data was not included in this report and agree this would have strengthened the conclusions. In this report, we aimed to consider VGSC expression more broadly as a class and to leverage the transcriptome level data available from large publicly available datasets. VGSC protein level data from these cohorts is not available, and VGSC do not appear on existing reverse phase proteome array (RPPA) panels. Moreover, high quality and specific antibodies are only available for a small number of the VGSC.  As noted in our response to Reviewer 1, however, we hope to examine the mechanism of VGSC effects in future reports. If we can elucidate cancer cells lines with different extremes of response to VGSC, either inherently or through siRNA screens, this may help us narrow down which VGSC are most closely linked to our associations. We agree that for these studies protein expression data will be essential. To address these concerns, we have added the following statement to the limitations noted in the discussion (lines 396-403):

“We also did not explore protein-level expression of VGSC. A protein biomarker which could be assessed by immunohistochemistry or Western blot would be much more useful for clinical application. Moreover, differences in mRNA may not always translate into protein expression differences. In future studies, we hope to assess the mechanism of VGSC inhibition in ovarian cancer more in-depth and plan to include protein level assessments in these investigations. "

Round 2

Reviewer 2 Report

The Authors have given

sufficient explanations regarding the commented 

expressed by the Reviewer.